# *C. elegans* neurons have functional dendritic spines

**Andrea Cuentas-Condori[1], Ben Mulcahy[2], Siwei He[3], Sierra Palumbos[3], Mei Zhen[2], David M Miller III[1,3]***

[1]Department of Cell and Developmental Biology, Vanderbilt University, Nashville, United States; [2]Lunenfeld-Tanenbaum Research Institute, University of Toronto, Toronto, Canada; [3]Neuroscience Program, Vanderbilt University, Nashville, United States

**Abstract** Dendritic spines are specialized postsynaptic structures that transduce presynaptic signals, are regulated by neural activity and correlated with learning and memory. Most studies of spine function have focused on the mammalian nervous system. However, spine-like protrusions have been reported in *C. elegans* (Philbrook et al., 2018), suggesting that the experimental advantages of smaller model organisms could be exploited to study the biology of dendritic spines. Here, we used super-resolution microscopy, electron microscopy, live-cell imaging and genetics to show that *C. elegans* motor neurons have functional dendritic spines that: (1) are structurally defined by a dynamic actin cytoskeleton; (2) appose presynaptic dense projections; (3) localize ER and ribosomes; (4) display calcium transients triggered by presynaptic activity and propagated by internal $Ca^{++}$ stores; (5) respond to activity-dependent signals that regulate spine density. These studies provide a solid foundation for a new experimental paradigm that exploits the power of *C. elegans* genetics and live-cell imaging for fundamental studies of dendritic spine morphogenesis and function.

DOI: https://doi.org/10.7554/eLife.47918.001

**\*For correspondence:**
david.miller@vanderbilt.edu

**Competing interests:** The authors declare that no competing interests exist.

## Introduction

The majority of excitatory synapses in the mammalian brain feature short, local protrusions from postsynaptic dendrites that respond to presynaptic neurotransmitter release (*Rochefort and Konnerth, 2012*). These dendritic 'spines' were originally described by Ramon y Cajal (*Yuste, 2015*) and are now recognized as key functional components of neural circuits. For example, spine morphology and density are regulated by neural activity in plastic responses that are strongly correlated with learning and memory (*Kozorovitskiy et al., 2005*; *Moser et al., 1997*). Although spine-like protrusions have been reported for invertebrate neurons (*Leiss, 2008*; *Petralia et al., 2016*), few studies (*Bushey et al., 2011*; *Scott et al., 2003*) have rigorously determined if these structures share functional features with vertebrate spines.

The anatomy of the *C. elegans* nervous system was originally defined by reconstruction of electron micrographs (EM) of serial sections. This approach revealed that a small subset of neurons displays short, spine-like protrusions. These include five classes of motor neurons (RMD, RME, SMD, DD, VD) and an interneuron (RIP) (*White et al., 1976*; *White et al., 1986*). Light and electron microscopy detected similar dendritic protrusions extending from motor neurons in the nematode, *Ascaris* (*Angstadt et al., 1989*; *Stretton et al., 1978*). Finally, recent reports used light microscopy to show that a postsynaptic acetylcholine receptor is localized near the tips of spine-like protrusions on the DD class motor neurons that directly appose presynaptic termini (*Oliver et al., 2018*; *Philbrook et al., 2018*). Here, we have adopted a systematic approach to demonstrate that spine-

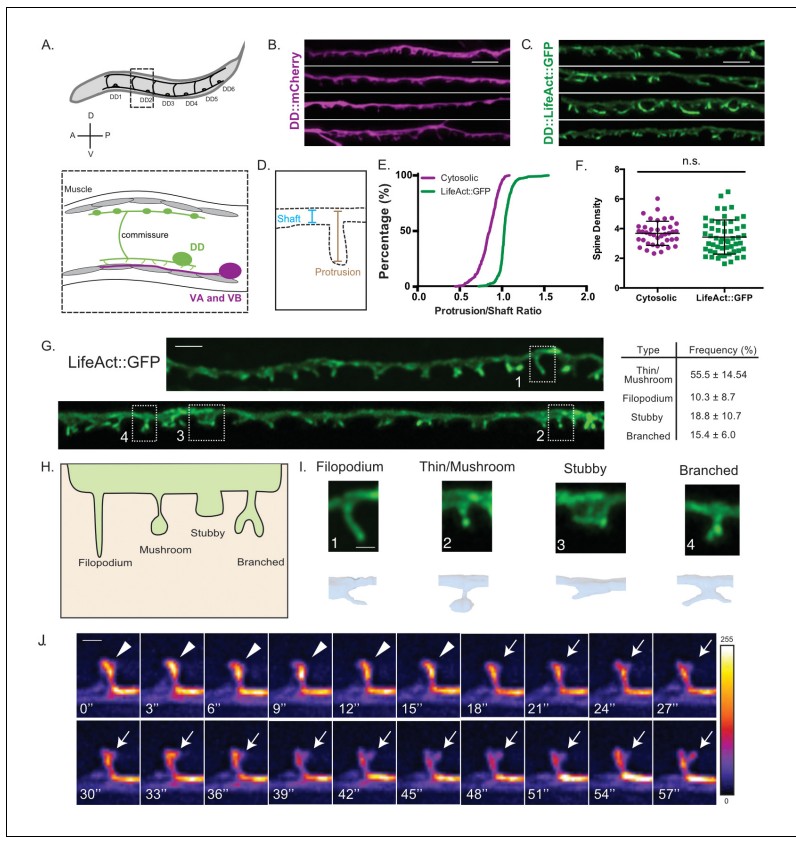

**Figure 1.** DD GABAergic neurons display dendritic spines. (**A**) Six DD motor neurons are located in the *C. elegans* ventral nerve cord. In the adult, DD presynaptic boutons (oblong ovals) innervate dorsal muscles (gray cells) and DD postsynaptic termini (spines) receive cholinergic input from VA and VB motor neurons on the ventral side (magenta). (**B–F**) Airyscan imaging resolves ventrally projecting spines from DD neurons labeled with (**B**) cytosolic mCherry or (**C**) LifeAct::GFP. (**D**) Intensity of spine over shaft ratio reveals that (**E**) LifeAct::GFP preferentially accumulates at the spine whereas cytosolic mCherry is evenly distributed between the spine and shaft (KS test, p<0.0001, n > 286 spines). (**F**) Spine density of young adults revealed by mCherry (3.68 ± 0.8 spines/ 10 μm) or LifeAct::GFP (3.43 ± 1.2 spines/10 μm) is not significantly different (t test, p=0.0855, n > 16 worms). Measurements are mean ± SD. Scale bars = 2 μm. (**G–I**) LifeAct::GFP reveals 1. Thin/Mushroom (55.5 ± 14.5%), 2. Filopodial (10.3 ± 8.70%), 3. Stubby (18.8 ± 10.4%), 4. Branched spines (15.4 ± 6.0%) in adult DD motor neurons. Measurements are mean ± SD, n = 16 worms, 357 spines. For scatterplot, see *Figure 1—figure supplement 1*. (Scale bar = 1 μm). (**H**) Schematic of spine shapes. (**I**) Images of each type of spine identified by (top) Airyscan imaging (Scale bar = 500 nm) of LifeAct::GFP or (bottom) 3D-reconstruction of DD1 from serial electron micrographs of a high-pressure frozen young adult. See also *Figure 1—figure supplement 1B*. (**J**) Dendritic spines are dynamic. Snapshots of in vivo spine remodeling from a thin/mushroom (arrowhead) to branched morphology (arrow). Images (LifeAct::GFP) are shown with a rainbow LUT. Higher intensity is represented by warm colors and dimmer intensity by cold colors. L4 stage larva. See also *Figure 1—figure supplement 1C-H*. Scale bar = 500 nm.

DOI: https://doi.org/10.7554/eLife.47918.002

The following source data and figure supplements are available for figure 1:

**Source data 1.** Individual measurements of spines/shaft intensity ratios labeled with cytosolic mCherry or LifeAct:: GFP.

DOI: https://doi.org/10.7554/eLife.47918.006

**Source data 2.** *Figure 1*_Density_by markers.

DOI: https://doi.org/10.7554/eLife.47918.007

**Figure supplement 1.** Dendritic spines adopt distinct morphologies.

DOI: https://doi.org/10.7554/eLife.47918.003

**Figure supplement 1—source data 1.** *Figure 5*_Spine_Density_by_genotype.

DOI: https://doi.org/10.7554/eLife.47918.004

**Figure supplement 2.** Dendritic spines display a dynamic actin cytoskeleton.

*Figure 1 continued on next page*

*Figure 1 continued*

DOI: https://doi.org/10.7554/eLife.47918.005

like structures in GABAergic motor neurons (DD and VD) exhibit the salient hallmarks of dendritic spines.

## Results and discussion

### Dendritic spines in *C. elegans* GABAergic neurons

In the adult, Dorsal D (DD) class GABAergic motor neurons extend axons to innervate dorsal muscles and receive cholinergic input at ventral dendrites (*Figure 1A*). We used Airyscan imaging, a type of super-resolution microscopy (*Korobchevskaya et al., 2017*), to detect spine-like projections on the ventral processes of adult DD neurons labeled with a cytosolic mCherry marker (*Figure 1B*). Because the actin cytoskeleton is a structural hallmark of vertebrate dendritic spines (*Cingolani and Goda, 2008*), we also labeled DD neurons with the actin marker, LifeAct::GFP (*Riedl et al., 2008*). Super-resolution images detected apparent enrichment of LifeAct::GFP in DD spines versus the dendritic shaft. For a quantitative assessment, we calculated the ratio of the spine to shaft fluorescence (*Figure 1D*) and plotted the cumulative distribution for each marker (*Figure 1E*). This representation shows a clear separation between measurements of cytosolic mCherry that is evenly distributed throughout dendrites (median spine/shaft ratio <1) versus that of the LifeAct::GFP signal (median spine/shaft ratio >1) (KS test, p<0.0001, *Figure 1E*). Thus, actin is enriched in DD spines. This interpretation is strengthened by our finding that independent measurements of the spine density (mean spine numbers/10 µm) with either cytoplasmic mCherry, LifeAct::GFP or the membrane bound marker, myristolated-mRuby (MYR::mRuby) are not significantly different (*Figure 1F* and *Figure 1—figure supplement 1A*). Similar results were obtained by EM reconstruction (see below) (*Figure 1—figure supplement 1A*).

A close inspection of LifeAct::GFP-labeled protrusions revealed a variety of spine shapes which we grouped into morphological classes resembling those previously reported for mammalian dendritic spines (*Bourne and Harris, 2008*; *Rochefort and Konnerth, 2012*): thin/mushroom, filopodial, stubby and branched (*Figure 1G–I*) (see Materials and methods). We merged 'thin' and 'mushroom' shapes into a single category because both are defined by an enlarged head region vs a narrower neck. By these criteria, adult DD neurons have predominantly thin/mushroom spines, with lesser fractions of filopodial, stubby and branched shapes (*Figure 1G* and *Figure 1—figure supplement 1B-D*). A comparison of the morphological classes identified with the LifeAct::GFP vs MYR::mRuby markers revealed some differences, notably the frequency of stubby spines, which is significantly elevated with the MYR::mRuby label (*Figure 1—figure supplement 1D*). These differences could reflect the relative ease of scoring different spine morphologies with markers for either the cell membrane (MYR::mRuby) or actin cytoskeleton

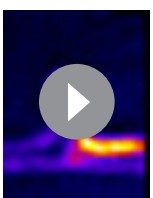

**Video 1.** In vivo spine dynamics. A thin spine extends a lateral projection to form a branched spine. Snapshots are displayed in *Figure 1J*. Pseudo-colored with Rainbow Dark LUT.

DOI: https://doi.org/10.7554/eLife.47918.009

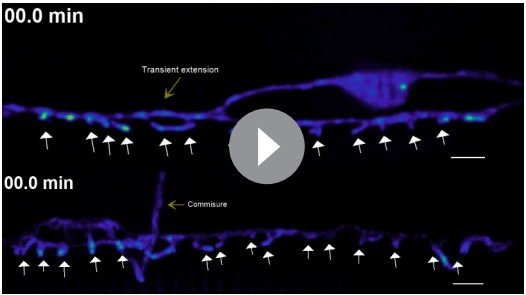

**Video 2.** LifeAct::GFP dynamics. LifeAct::GFP fluorescence at DD spines (white arrows) fluctuates between bright and dark signals. Asterisk labels DD cell body and yellow arrow labels transient protrusion and commissures. Scale bar = 5 µm. Pseudo-colored with Rainbow Dark LUT.

DOI: https://doi.org/10.7554/eLife.47918.010

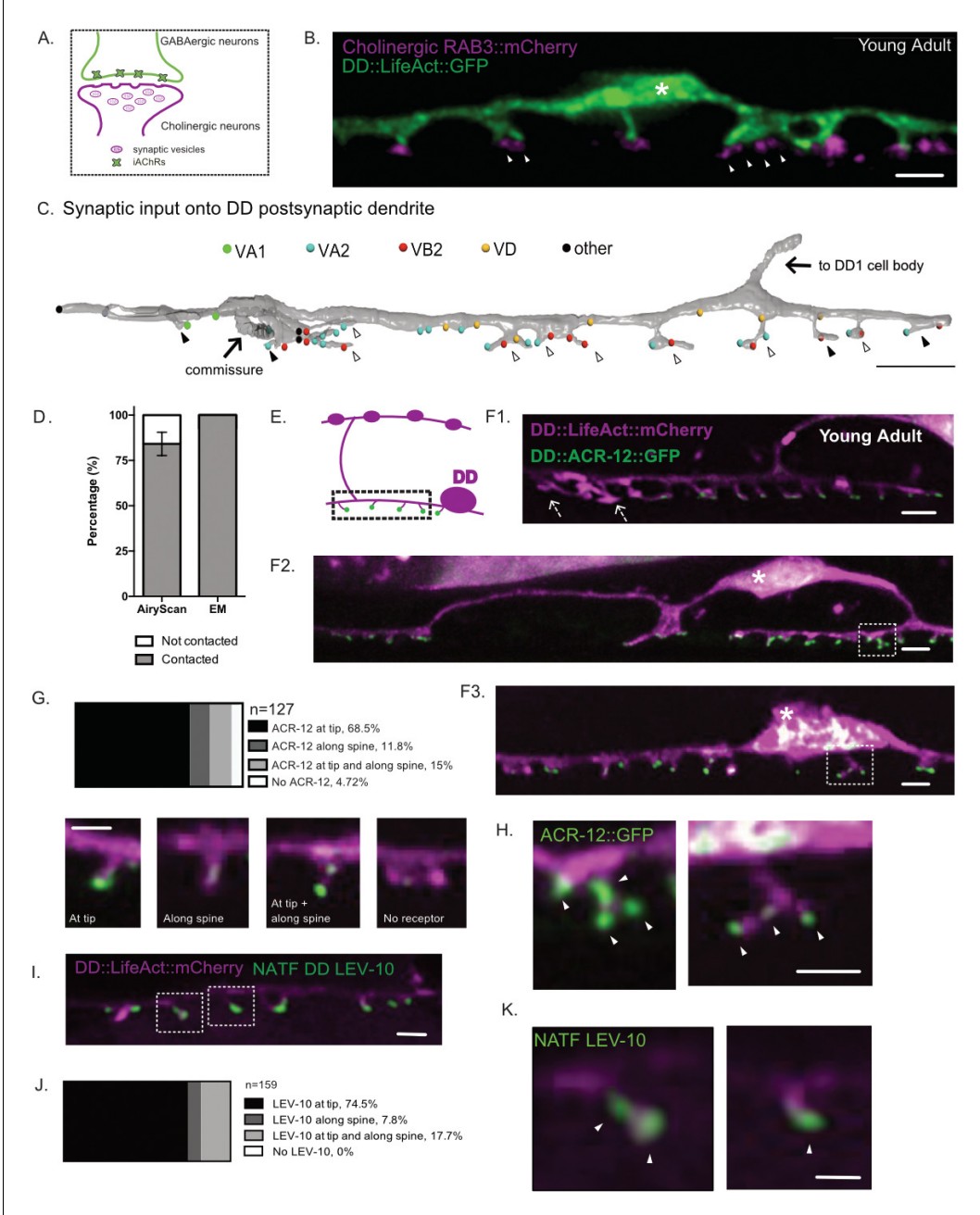

**Figure 2.** DD spines appose presynaptic cholinergic vesicles. (**A**) Postsynaptic ionotropic acetylcholine receptors (iAChRs) are localized in GABAergic motor neurons in apposition to input from cholinergic motor neurons. (**B**) Cholinergic RAB-3 presynaptic vesicles labeled with mCherry localize in close proximity to DD postsynaptic spines labeled with *flp-13*::LifeAct::GFP. Young adult stage worms. Scale bar = 1 μm. Asterisk denotes the DD cell body. Arrowheads mark multiple RAB-3::mCherry clusters apposing dendritic spines. (**C**) Volumetric EM reconstruction of a portion of the DD1 dendrite (25 μm) in the ventral nerve cord (gray) detects contacts with 43 presynaptic termini from axons of the cholinergic (VA1, VA2, VB2) and GABAergic (VD) motor neurons, and other neurons (other). 84.8% (n = 28/33) of VA and VB inputs are adjacent to DD spines. 33.3% (4/12) of spines directly oppose a single presynaptic partner (black arrowhead); the majority of spines (66.7%) appose more than one terminal (clear arrowhead). Scale bar = 2 μm. (**D**) Frequency of spines contacted by cholinergic presynaptic sites detected by Airyscan imaging (contacted, 84.1 ± 6.4% vs not contacted, 15.8 ± 6.4%, n = 128 spines from seven worms) or by EM (contacted, 100% vs not contacted, 0%, n = 12 spines from one worm). (**E**) Schematic of DD presynaptic boutons (top) and postsynaptic spines (dashed box) with distal iAChR puncta (green dots) on the ventral side. F1-3. iACh receptor subunit ACR-12::GFP (green) localizes to LifeAct::mCherry-labeled DD spines (magenta). Asterisk
*Figure 2 continued on next page*

*Figure 2 continued*

marks DD cell body. Arrows in F1denote spines without visible ACR-12::GFP clusters. Scale bars = 1 μm. (**G–H**) Locations of ACR-12::GFP puncta on DD spines. > 95% of spines have at least one ACR-12::GFP cluster (n = 127 spines from eight young adult worms). Examples of spines from each category. Scale bars = 1 μm. (**H**) Examples of spines with more than one ACR-12::GFP cluster. White arrowheads point to ACR-12::GFP clusters at DD spines. Scale bar = 200 nm. (**I–K**) NATF labeling of endogenous iAChR auxiliary protein LEV-10 in DD neurons (**I**) detects LEV-10 localization to spines with (**J**) all spines showing NATF LEV-10::GFP puncta (n = 159 spines from seven worms). Scale bars = 500 nm. (**K**) Example of spines with endogenous LEV-10 clusters. Scale bar = 200 nm.
DOI: https://doi.org/10.7554/eLife.47918.008

(LifeAct::GFP). Alternatively, over-expression of these markers could alter spine morphology but, in this case, does not appear to perturb overall spine density (*Figure 1—figure supplement 1A*).

As an independent method for assessing the presence of dendritic spines, we used EM to reconstruct the anterior-most 25 μm of the dendrite for DD1, the most anterior member of the DD class of motor neurons (*White et al., 1986*). For this experiment, young adult animals were prepared using High Pressure Freezing (HPF) to avoid potential artifacts arising from chemical fixation (*Mulcahy et al., 2018*; *Weimer et al., 2006*; *White et al., 1986*). Reconstruction of 50 nm serial ultrathin sections from the anterior DD1 dendrite detected twelve DD1 spines, with multiple morphological shapes (*Figure 2C*) that resemble classes revealed by fluorescent markers (*Figure 1I*; *Figure 1—figure supplement 1B-C*). EM reconstruction of mammalian dendritic spines in the hippocampus also revealed that thin and mushroom shapes predominate (*Harris and Stevens, 1989*; *Harris et al., 1992*) and that filopodial and stubby spines are less abundant (*Fiala et al., 1998*; *Zuo et al., 2005*).

## DD spines are shaped by a dynamic actin cytoskeleton

Although we have assigned DD motor neuron spines to four discrete classifications, both fluorescence imaging and EM reconstruction point to a broader array of spine types that includes potential intermediate forms (*Figure 1G–H*; 2C). A similarly heterogeneous array of spine shapes among mammalian neurons has been attributed to active remodeling of spine architecture (*Sala and Segal, 2014*; *Zuo et al., 2005*). To test for this possibility in *C. elegans*, we used live imaging to produce time-lapse recordings of DD spines. Our live-imaging revealed that some DD spines can remodel in vivo (*Video 1*). For example, *Figure 1J* depicts the emergence of a LifeAct::GFP-labeled nascent lateral branch near the tip of a thin/mushroom spine. During imaging sessions of ≥30 min, we observed cases of transient filopodial-like extensions (11 out of 25 movies) from the dendritic shaft that retract in the course of minutes (*Videos 2–3*). In contrast, most DD spines were stable throughout a given imaging session. In the mature mammalian cortex, extended imaging has revealed transient filopodial extensions with a lifetime shorter than a day, and potential plasticity over longer intervals, where approximately half of spines are stable for months (*Trachtenberg et al., 2002*; *Zuo et al., 2005*). To our knowledge, our time lapse images are the first to visualize dynamic dendritic spines in a motor neuron of a living organism (*Kanjhan et al., 2016b*).

Live-imaging of DD motor neurons also detected a dynamic actin cytoskeleton (*Figure 1—figure supplement 2A-E* and *Videos 2–3*), consistent with previous reports for mammalian dendritic spines (*Honkura et al., 2008*; *Mikhaylova et al., 2018*). To ask if actin assembly is required for DD spine morphogenesis (*Cingolani and Goda, 2008*), we applied genetic methods to knock down key regulators of actin polymerization and assessed their effect on DD spines (*Figure 1—figure supplement 2F*). We

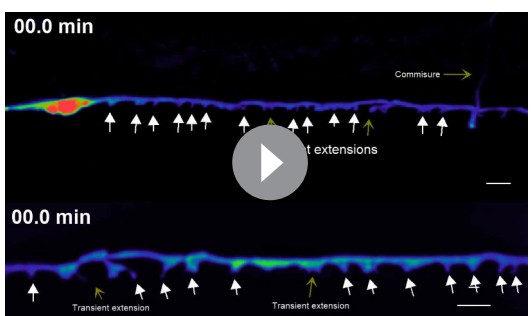

**Video 3.** Cytosolic mCherry dynamics. Cytosolic mCherry at DD spines (white arrows) shows modest changes in fluorescence over time. Yellow arrows label transient protrusions and commissures. Scale bar = 5 μm. Pseudo-colored with Rainbow Dark LUT.
DOI: https://doi.org/10.7554/eLife.47918.011

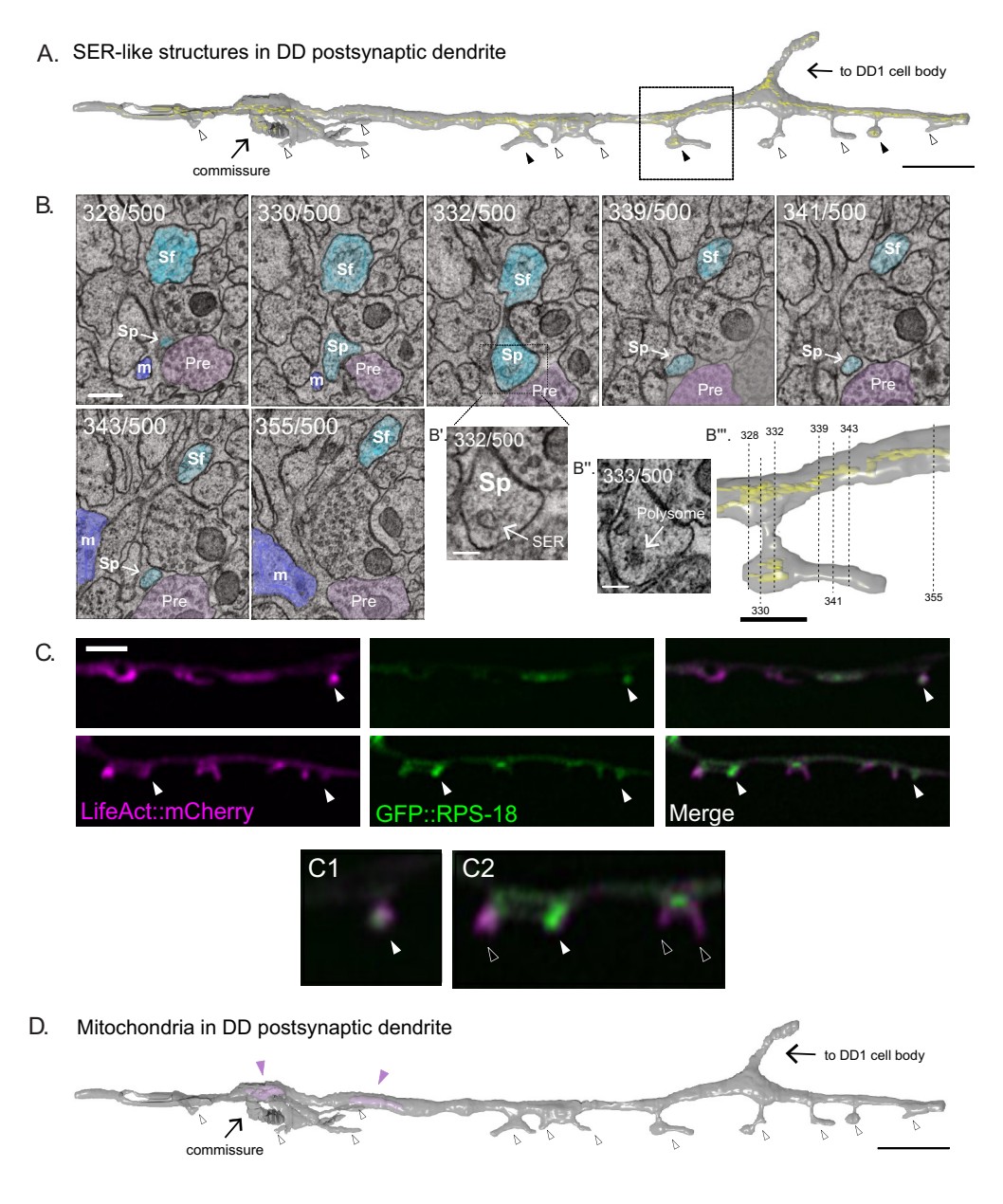

**Figure 3.** SER-like structures and ribosomes in spines and dendritic shaft. (**A**) 3D EM reconstruction of DD1 dendrite reveals Smooth Endoplasmic Reticulum (SER)-like cisternae (yellow) in the dendritic shaft and some spines (black arrowheads). Most spines lack SER-like structures (clear arrowheads). Scale bar = 2 μm. (**B**) Serial cross-sections (328-355) of the ventral nerve cord show spines (Sp) budding from DD1 (blue) dendritic shaft (Sf). 'Pre' labels presynaptic terminals from a cholinergic VA neuron (pink); m, muscle arm (purple). (Scale bar = 200 nm) (**B'**) Magnified region of section 332. Arrow points to SER-like structure in DD dendritic spine (Scale bar = 100 nm). (**B''**) Section 333. Arrow points to polysome-like structure in DD dendritic spine (Scale bar = 100 nm). (**B'''**) Volumetric reconstruction of DD1 dendrite (gray) and SER-like structures (yellow). Dashed lines denote location of each section shown in B. Scale bar = 500 nm. (**C**) Airyscan imaging shows GFP-labeled ribosomal protein, RPS-18 (***Noma et al., 2017***), localized to DD spines (arrowheads) labeled with LifeAct::mCherry. Scale bar = 2 μm. (**D**) Volumetric EM reconstruction of a portion of the DD1 (25 μm) dendrite shows mitochondria (purple) in the shaft (arrowheads) but not in spines (clear arrowheads).

DOI: https://doi.org/10.7554/eLife.47918.013

found that the Arp2/3 complex, and two of its activators, the F-BAR protein TOCA-1 (*Ho et al., 2004*) and the Wave Regulatory Complex (*Chen et al., 2010*), are required to maintain DD spine density (*Figure 1—figure supplement 2F*). Restoring expression of TOCA-1 to DD neurons rescued the spine density defect (*Figure 1—figure supplement 2G*), demonstrating that actin polymerization is required cell-autonomously to promote spine formation. Disruption of the Arp2/3 complex or its activators has been previously shown to reduce dendritic spine density in the mammalian brain and to impair function (*Kim et al., 2013*; *Lippi et al., 2011*; *Soderling et al., 2003*; *Spence et al., 2016*; *Westphal et al., 2000*).

## Dendritic spines of DD neurons directly appose presynaptic terminals

Because functional dendritic spines are sites of presynaptic input (*Alvarez and Sabatini, 2007*; *Hering and Sheng, 2001*; *Petralia et al., 2016*), we investigated the disposition of DD spines vis-a-vis their main presynaptic partners, the cholinergic VA and VB class motor neurons (*Figure 2A*). For super-resolution imaging, we used the synaptic vesicle-associated marker, mCherry::RAB-3, to label VA and VB presynaptic termini and LifeAct::GFP to label DD neurites and spines (*Figure 2B*). Clusters of mCherry::RAB-3-labeled puncta are located adjacent to DD spines (*Figure 2B*) (*Philbrook et al., 2018*). Among the 128 spines identified by LifeAct::GFP, most (~84%) reside near at least one presynaptic cluster (denoted 'contacted' in *Figure 2D*). Approximately ~40% (51/128) of DD spines are associated with multiple presynaptic clusters of RAB-3 puncta which suggests that individual spines receive input from >1 presynaptic terminal (arrowheads, *Figure 2B*).

Our EM reconstruction of a segment of DD1 dendrite revealed 12 spines, all in direct apposition with the presynaptic termini of cholinergic motor neurons (VA1, VA2 and VB2) (*Figure 2C–D*). Of the 33 cholinergic presynaptic inputs in this region, 84.8% (n = 28/33) appose DD1 spines, whereas only 15.2% (n = 5/33) are positioned along the dendritic shaft. This finding parallels the observation that only 10% of excitatory synapses in the mature mammalian cortex are positioned on dendritic processes (*Cingolani and Goda, 2008*). Two thirds of DD1 spines (n = 8/12) receive input from more than one neuron class. That is, a single DD1 spine head is contacted by presynaptic termini of both VA and VB class cholinergic motor neurons (*Figure 2C* and *Video 4*). This finding could explain the observation above from Airyscan imaging that multiple mCherry::RAB-3 puncta are adjacent to ~40% of DD spines (*Figure 2B*, arrowheads). We note that individual spines on GABAergic neurons in the mammalian hippocampus can also have inputs from multiple presynaptic termini (*Acsády et al., 1998*; *Gulyás et al., 1992*; *Petralia et al., 2016*). Interestingly, the DD1 dendrite also receives a few inhibitory inputs from the other class of GABAergic motor neurons (VD1), but most are restricted to the DD1 dendritic shaft (n = 5/6) (*Figure 2C*).

The acetylcholine receptor (AChR) subunit ACR-12 is postsynaptic to cholinergic inputs at GABAergic motor neurons (*Petrash et al., 2013*), and has been previously shown to localize to DD1 dendritic protrusions (*Figure 2E*) (*He et al., 2015*; *Philbrook et al., 2018*). We used Airyscan imaging to quantify the subcellular distribution of the ACR-12::GFP signal on DD spines. We detected ACR-12::GFP clusters on ~95% of DD spines (n = 121/127) (*Figure 2F–G*), with 68.5% (n = 87/127) localized at spine tips and the remainder either positioned along the lateral side

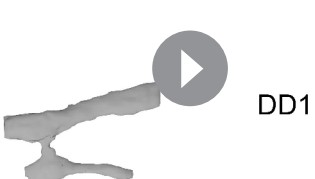

DD1

**Video 4.** DD1 dendritic spine receives input from cholinergic neurons. 3D EM reconstruction shows that a single DD spine (gray) contacts presynaptic terminals of VA (blue) and VB (pink) neuron. Muscle arms labeled in green.
DOI: https://doi.org/10.7554/eLife.47918.012

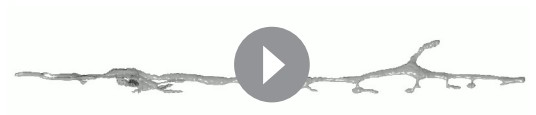

**Video 5.** 3D EM reconstruction of DD1 dendrite. DD1 dendrite (gray) has dendritic spines that contain ER (yellow) but no mitochondria (pink).
DOI: https://doi.org/10.7554/eLife.47918.014

of the spine (~12%, n = 15/127) or at both the side and tip (~15%, n = 19/127) (*Figure 2G*). 43.3% of spines (n = 55/127) had more than one ACR-12::GFP cluster (*Figure 2H*), a finding that mirrors a recent observation that spines of mammalian cortical neurons can display multiple assemblages of the postsynaptic protein PSD-95 (*Hruska et al., 2018*).

To obviate the possibility that localization of ACR-12::GFP clusters to DD spines results from over-expression, we used a new live-cell labeling scheme to detect an endogenous component of the acetylcholine receptor complex, the co-factor protein LEV-10 (*Gally et al., 2004*). NATF (Native and Tissue-Specific Fluorescence) relies on the reconstitution of superfolder GFP (sfGFP) from the split-sfGFP fragments, GFP1-10 and GFP11 (*He et al., 2019*). We used genome editing to fuse seven tandem copies of GFP11 to the C-terminus of the native LEV-10 coding sequence. When GFP1-10 was selectively expressed in DD neurons from a transgenic array (i.e., *Pflp-13::GFP1-10*), we detected LEV-10 NATF-GFP signal at 100% (n = 159/159) of DD spines (*Figure 2I–K*), further substantiating the idea that DD spines are sites of presynaptic input.

Our EM analysis confirmed that DD spines do not display electron dense postsynaptic densities (PSDs), a feature that is also not detected in electron micrographs of most *C. elegans* postsynaptic terminals (*Lim et al., 2016*; *White et al., 1976*; *White et al., 1986*; *Zhen et al., 2000*). Although robustly stained PSDs are observed at vertebrate glutamatergic synapses, PSDs are either absent or much less prominent in electron micrographs of vertebrate synapses at spines for other neurotransmitters (glycine, GABA, acetylcholine) (*Knott et al., 2002*; *Kubota et al., 2007*; *Umbriaco et al., 1994*). Postsynaptic assemblages at these synapses likely comprise distinct sets of scaffolding proteins, some of which are readily stained by heavy atoms used for EM imaging of glutamatergic synapses (*Petralia et al., 2005*; *Petralia et al., 2016*).

## ER and ribosomes localize to DD neuron dendritic spines

Key cytoplasmic organelles such as Smooth Endoplasmic Reticulum (SER), are present in both dendritic shafts and spines of mammalian neurons (*Harris and Stevens, 1989*). In addition to its role of processing membrane proteins, spine SER regulates activity-dependent $Ca^{++}$ release through the ryanodine receptor (*Fill and Copello, 2002*). Other structures such as polysomes and rough ER have also been reported in spines, consistent with the possibility of local translation at synapses (*Bourne and Harris, 2008*; *Hafner et al., 2019*; *Nimchinsky et al., 2002*; *Steward and Reeves, 1988*).

Our EM reconstruction of DD1 revealed cisternae SER-like structures in both the dendritic shaft and spines of DD1 (*Figure 3A–B*) and apparent ribosomes in some DD1 spines (*Figure 3B''*).

Consistent with this observation are our light microscopy images that detect the ribosomal protein RPS-18::GFP (*Noma et al., 2017*) in about half of (44.5 ± 12.0%) DD spines (*Figure 3C*). Mitochondria and microtubules are reported to be rare in the dendritic spines of mature mammalian neurons (*Bourne and Harris, 2008*; *Nimchinsky et al., 2002*). Our EM reconstruction did not find mitochondria or microtubules in all twelve DD spines (*Figure 3D*), while both organelles were detected in the DD1 dendritic shaft (*Video 5*). These observations however should be interpreted cautiously given the small number of spines reconstructed.

## Activation of presynaptic cholinergic motor neurons drives $ca^{++}$ transients in DD spines

$Ca^{++}$ is a key signaling molecule to mediate activity-dependent synaptic plasticity (*Lee et al., 2016*; *Rochefort and Konnerth, 2012*). We reasoned that functional DD spines should exhibit dynamic $Ca^{++}$ transients. To test this hypothesis, we expressed the $Ca^{++}$ sensor GCaMP6s in DD neurons. Live-imaging (at 2 s intervals) revealed spontaneous $Ca^{++}$ transients in both DD spines and shafts that lasted for several seconds (*Figure 4A–C* and *Video 6*). Muscle cells that share cholinergic input with DD spines also display prolonged elevation of $Ca^{++}$ over a period of seconds (*Liu et al., 2013*), suggesting that cholinergic neurons regulate the long lasting bursts of $Ca^{++}$ in spines and muscle cells.

Interestingly, $Ca^{++}$ transients were observed simultaneously in adjacent spines about 50% of the time (*Figure 4D*). To estimate the likelihood of simultaneous $Ca^{++}$ peaks occurring by chance, we compared the distribution of the observed time differences between neighboring spine $Ca^{++}$ peaks (△T) to that of a uniform distribution (at 2 s intervals) over the period of observation (20 s). These

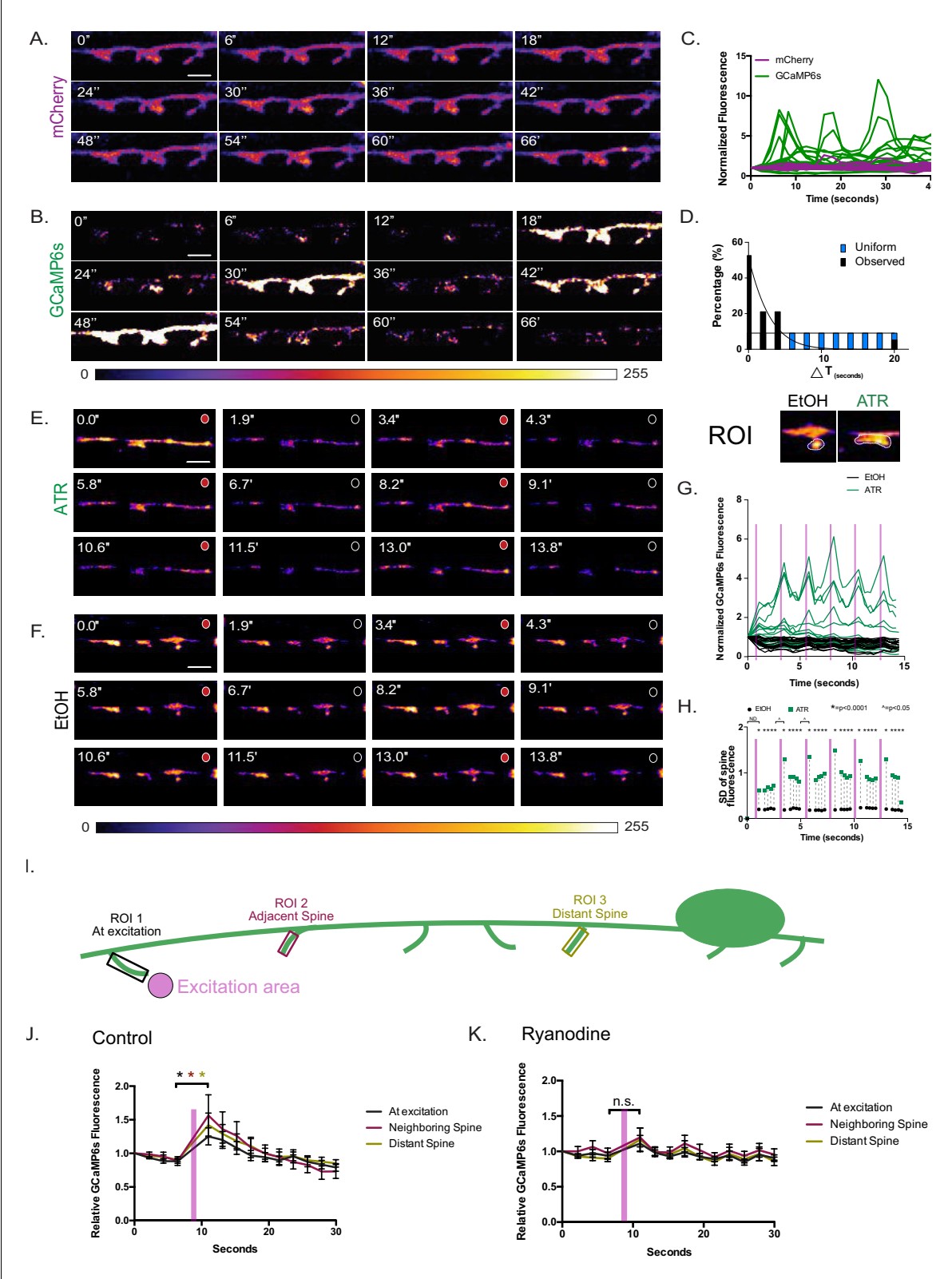

**Figure 4.** Coordinated Ca$^{++}$ transients in dendritic spines. Series (time in seconds) of live-cell images of cytosolic (A) mCherry and (B) GCaMP6s in DD postsynaptic spines reveals (C) dynamic GCaMP6s vs stable mCherry signals, n = 11 movies, 31 spines. (D) GCaMP6s transients occur in neighboring spines more frequently (>50%) than predicted by a random distribution (KS test, p<0.0001). Scale bars = 500 nm. (E–H) VA motor neuron activation is correlated with Ca$^{++}$ transients in DD1 spines. GCaMP6s fluorescence imaged (at 0.5 s intervals) with periodic optogenetic activation (at 2.5 s intervals)

*Figure 4 continued on next page*

*Figure 4 continued*

of ceChrimson, detects Ca$^{++}$ transients with (**E**) ATR (n = 14) but not with carrier (**F**) (EtOH) (n = 12). Circles at the top right corner of each panel correspond to red light on (red) for ceChrimson activation vs off (black). Scale bars = 500 nm. (**G**) GCaMP6s fluorescence throughout the 15 s recording period plotted for ATR (green) (n = 14) vs carrier (EtOH) (black) (n = 12). (**H**) Plot of the standard deviation (SD) of GCaMP6s signal at each time-point shows that fluctuations in the ATR-treated samples (green boxes) are significantly greater than in EtOH controls (black circles), F-test, *=p < 0.0001. Additionally, SDs are significantly different between timepoints before and after light activation (T$_6$ vs T$_7$ and T$_{11}$ vs. T$_{12}$). F-test, ^=p < 0.05. ND = not determined. Purple bars denote interval with red-light illumination (e.g., ceChrimson activation). (**I–K**) Ca$^{++}$ propagation to neighboring spines depends on intracellular Ca$^{++}$ stores. (**I**) Graphical representation of the experimental paradigm: 561 nm laser excitation at a single spine (excitation area, pink) with subsequent GCaMP6s changes recorded from three different regions of interest at: (1) the excited spine, (2) an adjacent spine and (3) a distant spine. (**J**) In the wild type, significant Ca$^{++}$ changes are detected at the excited spine (p=0.0182), adjacent spine (p=0.0319) and distant spine (p=0.0402), n = 15 videos. (**K**) In ryanodine-treated worms, significant Ca$^{++}$ changes are not detected at the excited spine (p>0.999), at an adjacent spine (p=0.924) or at a distant spine (p=0.552), n = 16 videos.
DOI: https://doi.org/10.7554/eLife.47918.015

The following source data and figure supplement are available for figure 4:

**Source data 1.** *Figure 1*_TipShaft Ratio.
DOI: https://doi.org/10.7554/eLife.47918.017
**Source data 2.** *Figure 4*_DirectedCa_All traces.
DOI: https://doi.org/10.7554/eLife.47918.018
**Figure supplement 1.** Dendritic Ca$^{++}$ transients depend on intracellular Ca$^{++}$ stores.
DOI: https://doi.org/10.7554/eLife.47918.016

distributions are statistically different (KS test, p<0.0001), suggesting that Ca$^{++}$ dynamics may reflect mechanisms for postsynaptic coordination of adjacent DD spine activity (*Figure 4D*). Alternatively, coordinated Ca$^{++}$ transients in adjacent spines could arise from shared presynaptic neuron inputs. For example, the majority of DD1 spines (9/12) are postsynaptic to the same cholinergic motor neuron VA2 (*Figure 2C*). A similar explanation of convergent input was proposed for the coordinated firing of adjacent dendritic spines in rat hippocampal neurons (*Takahashi et al., 2012*).

We did not observe Ca$^{++}$ signals in DD spines when cholinergic receptors were desensitized by administration of an agonist levamisole (data not shown). This finding is consistent with the hypothesis that Ca$^{++}$ transients in DD spines depend on presynaptic cholinergic signaling. To test this idea, we engineered a transgenic animal for optogenetic activation of VA neurons with ceChrimson (*Punc-4::ceChrimson::SL2::3xNLS::GFP*) (*Schild and Glauser, 2015*) and detection of Ca$^{++}$ changes in DD spines with GCaMP6 (*Pflp-13*::GCaMP6s::SL2::mCherry). VA motor neurons are presynaptic to DDs and therefore are predicted to evoke DD neuronal activity (*Philbrook et al., 2018*; *White et al., 1986*). ceChrimson was activated by a brief flash of 561 nm light (80 ms) at 2.5 s intervals and the GCaMP6s signal in DD neurons was recorded at 2 Hz. This experiment detected a correlation of GCaMP6s fluorescence with ceChrimson activation (*Figure 4E–G* and *Videos 7–8*). Although GCaMP6s fluorescence also varied in DD spines in the absence of 561 nm illumination, fluctuations were strongly correlated with ceChrimson activation as shown by a plot of the standard deviation of GCaMP6s fluorescence for all traces across the 15 s sampling period (*Figure 4H*). These results are consistent with the interpretation that DD spines are responding to cholinergic input from presynaptic VA motor neurons and parallel an earlier observation that DD dendritic protrusions are required for cholinergic activation of Ca$^{++}$ transients in DD cell soma (*Philbrook et al., 2018*).

Activation of ceChrimson triggered Ca$^{++}$ changes at neighboring spines and in the DD shaft (*Figure 4F*). As noted above, this effect could arise from shared input to adjacent spines from a single presynaptic motor neuron (*Figure 2C*). Ca$^{++}$ waves might also spread

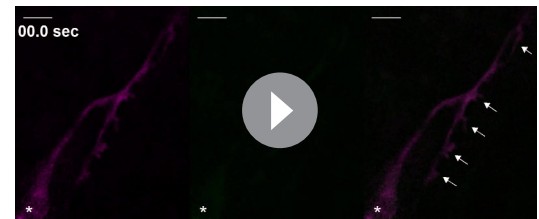

**Video 6.** Intrinsic Ca$^{++}$ waves in DD spines. GCaMP6s (green, center panel) detects spontaneous Ca$^{++}$ transients in the DD1 dendritic shaft and spines (arrows). Note the relatively constant cytosolic mCherry signal (magenta, left panel) GCaMP6s and mCherry signals are merged in the right panel. Scale bar = 2 μm. Pseudo-colored with Rainbow Dark LUT.
DOI: https://doi.org/10.7554/eLife.47918.019

**Video 7.** Cholinergic activation triggers Ca$^{++}$ transients in DD spines. Three examples of GCaMP6s changes recorded with activation of cholinergic neurons every 2.5 s (pink circle) on ATR-grown worms. Scale bar = 2 μm. Pseudo-colored with Rainbow Dark LUT. Arrows denote spines.
DOI: https://doi.org/10.7554/eLife.47918.020

**Video 8.** Cholinergic activation does not trigger large changes in cytosolic Ca$^{++}$ in DD spines in the absence of ATR. Three examples of GCaMP6s changes recorded after activation of cholinergic neurons every 2.5 s (pink circle) in control worms grown in the absence of ATR. Scale bar = 2 μm. Pseudo-colored with Rainbow Dark LUT. Arrows denote spines. Asterisk (left panel) marks DD neuron soma.
DOI: https://doi.org/10.7554/eLife.47918.021

along the DD dendrite from a locally activated spine. To test for this possibility, we restricted ceChrimson activation to regions adjacent to single DD spines and recorded Ca$^{++}$ changes (1) at the excited spine; (2) at an adjacent spine and (3) at a distant spine (See Materials and methods) (*Figure 4I*). Live imaging two seconds after ceChrimson activation detected elevated GCaMP signals in spines in all three regions that then waned with time (*Figure 4J* and *Video 9*). This observation suggests that local cholinergic release can trigger Ca$^{++}$ changes in neighboring and distant spines. In developing hippocampal neurons, an initial Ca$^{++}$ transient can be propagated to neighboring spines via Ca$^{++}$ release from intracellular stores (*Lee et al., 2016*). Because activation of intracellular Ca$^{++}$ stores depends on ryanodine-sensitive channels, we repeated the local spine activation experiment (*Figure 4I*) in the presence of 1 mM ryanodine. Blocking ryanodine receptor-dependent Ca$^{++}$ release substantially attenuated Ca$^{++}$ transients in all spines (*Figure 4K*) thus suggesting that intracellular Ca$^{++}$ is required to propagate the initial activation. In hippocampal neurons, local glutamate uncaging triggers ryanodine-insensitive Ca$^{++}$ transients in the excited spine but this effect is rapid (~250 ms) (*Lee et al., 2016*) and thus a similar rapid initial Ca$^{++}$ transient might have been missed by our experimental set up that detects GCaMP signals ~ 2 s after ceChrimson activation (*Figure 4K*). Our finding of prominent SER-like structures in the DD shaft and spines is consistent with the idea that Ca$^{++}$ release from intracellular stores could drive coordinated activation of adjacent spines (*Figure 3A–B*). Further, a mutation that disrupts the UNC-68/Ryanodine receptor function results in substantially reduced intrinsic Ca$^{++}$ dynamics in DD spines (*Figure 4—figure supplement 1A*). Together, our results suggest that Ca$^{++}$ signals can propagate to nearby spines after local activation and that spreading depends on intracellular Ca$^{++}$ stores.

## Cholinergic signaling enhances DD spine density during development

In mammalian neurons, spine shape and density are modulated throughout development (*Fiala et al., 1998*; *Harris et al., 1992*; *Kanjhan et al., 2016a*). LifeAct::GFP imaging in late larval to adult stages (L3, early L4, mid-L4 and young adult), revealed that spine density increases as DD neurons elongate during development (*Figure 5B and E* and S4A).

Dendritic spines can be modulated by changes in synaptic strength (*Bourne and Harris, 2008*; *Nimchinsky et al., 2002*; *Rochefort and Konnerth, 2012*). Long-term

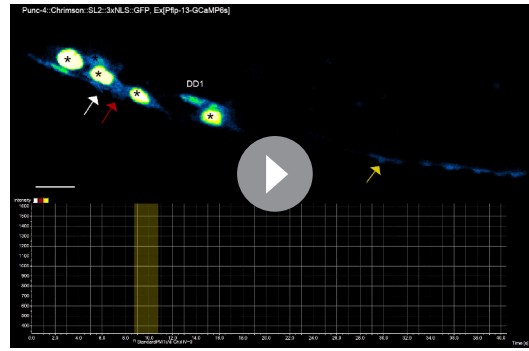

**Video 9.** Local cholinergic activation triggers Ca$^{++}$ transients in neighboring spines. Top: Activation of cholinergic neurons at a single spine (white arrow) triggers Ca$^{++}$ changes at adjacent (red arrow) and distant spines (yellow). ceChrimson::SL2::3xNLS:GFP labels DA and VA nuclei in green (black asterisks). The DD1 cell soma is labeled. Scale bar = 5 μm. Bottom: Traces of local Ca$^{++}$ changes at excited (red), adjacent (white) and distant (yellow) spines over time. Vertical yellow bar denotes the 561 nm excitation window. Note that GCaMP6s signals during the period of 561 nm excitation are interpolated from signals measured immediately before and after excitation.
DOI: https://doi.org/10.7554/eLife.47918.022

potentiation is correlated with increased numbers of spines in the mammalian brain (*Engert and Bonhoeffer, 1999*; *Trachtenberg et al., 2002*; *Holtmaat et al., 2005*). Similarly, hyperactivity is associated with increased dendritic spine density in hypoglossal motor neurons (*Kanjhan et al., 2016b*). Conversely, long-term depression induces spine shrinkage (*Zhou et al., 2004*) and may lead to their elimination (*Hasegawa et al., 2015*).

We tested whether DD spines respond to changes in cholinergic input by altering acetylcholine levels. To reduce cholinergic signaling, we used *unc-17/vAChT (e113)* mutants, in which expression of the vesicular acetylcholine transporter vAChT is selectively eliminated in ventral cord cholinergic motor neurons (J. Rand, personal communication). Conversely, to elevate cholinergic signaling, we used *ace-1(p100);ace-2(g720)* mutants that disrupt acetylcholinesterase activity (*Figure 5A*). Reduced acetylcholine release in the *unc-17/vAChT* mutant results in reduced DD spine density at the L4 larval stage (*Figure 5C and E* and S4B). In contrast, increased acetylcholine levels (i.e., in *ace-1;ace-2* mutants) results in precocious elevation of spine density during development (*Figure 5D and E* and S4C). Importantly, either chronic reduction (i.e., in the *unc-17/vAChT* mutant) or elevation (i.e., in *ace-1;ace-2* double mutants) of acetylcholine levels (*Figure 5—figure supplement 1B and C*) impaired the developmentally regulated enhancement of spine density that is normally observed in wild-type animals. These findings are consistent with the idea that cholinergic signaling positively regulates the formation of DD spines. The developmental elevation of spine density is also blocked in *unc-31/CAPS* mutants (*Speese et al., 2007*), in which neuropeptide and catecholamine neurotransmitter release is also prevented (*Figure 5—figure supplement 1D and E*).

For an additional test of activity-dependent regulation of DD spine density, we modulated presynaptic cholinergic function for specific periods during larval development. To reduce cholinergic activity, we expressed the histamine-gated chloride channel (*Pokala et al., 2014*) in A-class (DA, VA) cholinergic motor neurons, presynaptic partners of DD neurons (*Figure 2C*) (*White et al., 1976*). Animals grown in the presence of histamine showed reduced spine density at the L4 stage compared to animals grown on plates without histamine (*Figure 5F–I*). To elevate cholinergic activity, we expressed ceChrimson (*Schild and Glauser, 2015*) in A-class motor neurons and measured spine density at the L3 stage when wild-type animals show fewer DD spines than in adults (*Figure 5B and E*). Animals were exposed to 561 nm light to activate ceChrimson for a brief period (1 s every 4 s) during the L2-L3 stage larval development (See Materials and methods). This treatment led to increased spine density (scored in L3 larvae) in comparison to animals without ceChrimson stimulation (*Figure 5J–M*). These results demonstrate that DD spine density depends on presynaptic cholinergic signaling. Thus DD spines share the property of mammalian dendritic spines of positive regulation by neuronal activity (*Kanjhan et al., 2016a*; *Holtmaat et al., 2005*).

## VD-class GABAergic neurons also display dendritic spines

In the adult *C. elegans* motor circuit, dendrites of the DD-class GABAergic motor neurons receive cholinergic input in the ventral nerve cord, whereas the VD class receives input in the dorsal nerve cord (*Figure 4—figure supplement 1A*). Because the original EM reconstruction of the *C. elegans* nervous system detected spine-like structures on VD neurons (*White et al., 1976*), we sought to verify this finding by using the LifeAct::GFP marker for Airyscan imaging. We used miniSOG (*Qi et al., 2012*) for selective ablation of DDs (See Materials and methods) since the LifeAct::GFP marker (*Punc-25*::LifeAct::GFP), in this case, was expressed in both DD and VD neurons. This experiment confirmed the presence of dendritic spines in VD neurons throughout the dorsal nerve cord (*Figure 5—figure supplement 2A-D*).

Our EM reconstruction of 27 µm of the anterior VD2 dendrite detected nine dendritic spines (*Figure 4—figure supplement 1E*). Similar to DD1, most VD spines are juxtaposed to presynaptic termini of cholinergic motor neurons (DA2, DB1, AS2). Additional presynaptic inputs from other cholinergic and GABAergic motor neurons (DD1) are distributed along the dendritic shaft (*Figure 4—figure supplement 1F*). Several mitochondria are also observed in VD2 dendritic shaft (*Figure 4—figure supplement 1E*). Thus, both the DD and VD classes of ventral cord GABAergic motor neurons display dendritic spines (*White et al., 1976*; *White et al., 1986*).

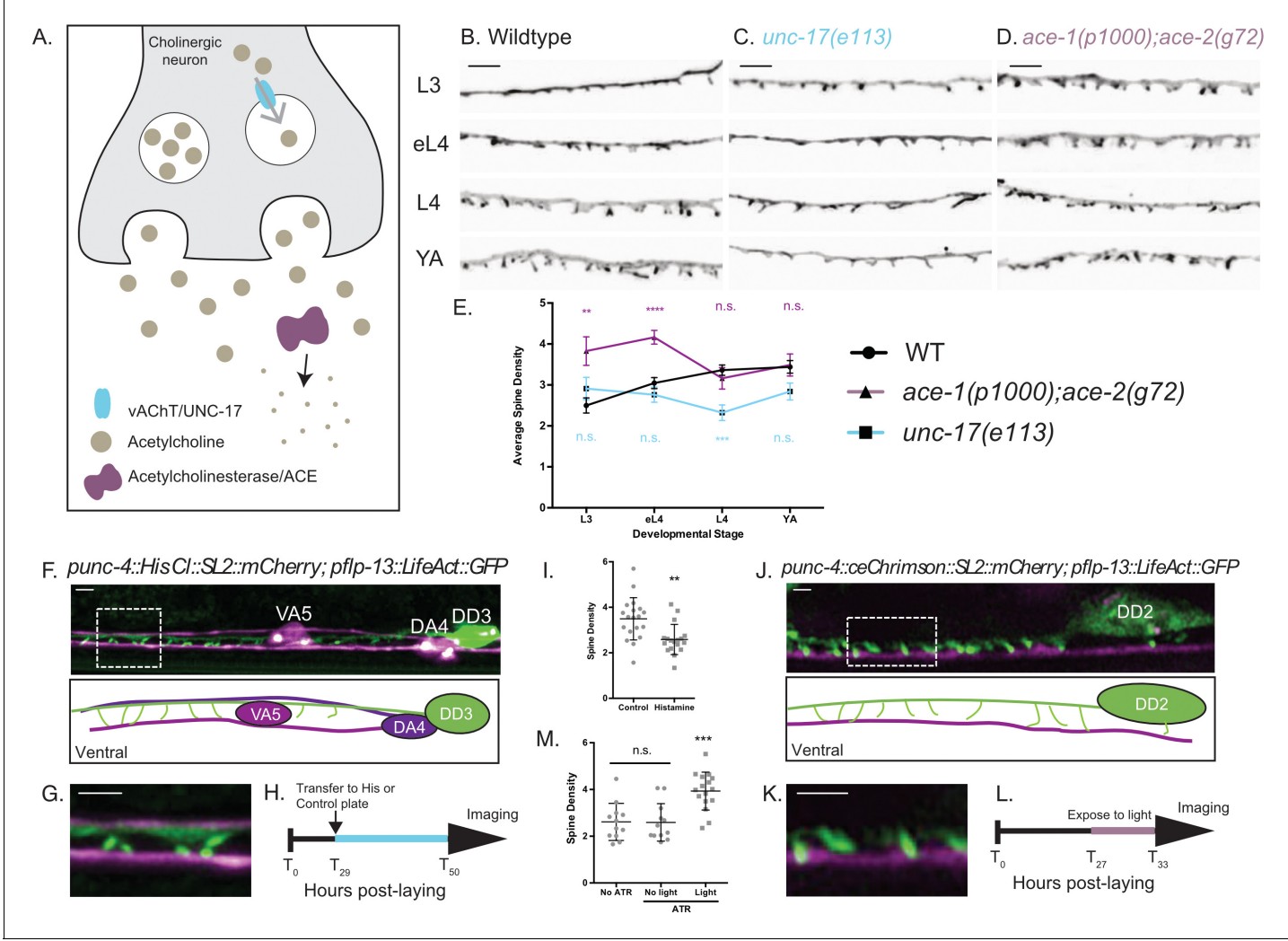

**Figure 5.** Cholinergic activity regulates spine density during development. (**A**) Synaptic vesicles are loaded with acetylcholine (ACh) by the vesicular acetylcholine transporter (vAChT/UNC-17). Acetylcholinesterase enzymes (ACE) degrade synaptic ACh. (**B–E**) Spine density increases throughout development in the wild type (WT) but not in *unc-17(e113)* mutants whereas spine density is precociously elevated in *ace-1(p1000);ace-2(g72)* mutants. Representative images of (**B**) WT, (**C**) *unc-17 (e113)* and (**D**) *ace-1(p100); ace-2(g72)*. Scale bars = 2 μm. See *Figure 5—figure supplement 1* for scatter plots for (**E**). (**F–I**) Reduced ACh signaling in cholinergic motor neurons decreases postsynaptic spine density. (**F**) Expression of Histamine-gated Chloride channels and mCherry in cholinergic (VA and DA) motor neurons (*punc4::HisCl::SL2::mCherry*) (*Pokala et al., 2014*) vs DD motor neurons labeled with LifeAct::GFP shows (**G**) DD spines (green) extending to the ventral process of the VA5 motor neuron (magenta). Note dorsal placement of DA4 axon. (**H**) Synchronized L2 larvae were transferred to either histamine or control plates at $T_{29}$ (hours post-laying) for growth up to the L4 stage (~$T_{50}$), See Materials and methods. (**I**) DD spine density at the L4 stage is reduced by growth on histamine (2.58 ± 0.6) vs control (3.49 ± 0.9). T-test, **=p < 0.01, n > 17. Scale bars = 1 μm. (**J–M**) Temporal activation of A-class cholinergic motor neurons increases spine density. (**J**) Cholinergic motor neurons (e.g., VA4) express ceChrimson (*Schild and Glauser, 2015*) and mCherry (*punc4::ceChrimson::SL2::mCherry*). LifeAct::GFP marks DD2. (**K**) DD spines (green) extend ventrally toward VA process (magenta). (**L**) Synchronized L2 stage larvae ($T_{27}$, hours post-laying) were transferred to ATR or control plates (see Materials and methods) for 6 hr (until $T_{33}$,~L3 stage) and exposed to red-light pulses vs control group grown in the dark. (**M**) Exposure to red-light for 6 hr elevates spine density at the L3 stage (3.9 ± 0.8) vs control (2.59 ± 0.8). T-test, *** is p<0.001, n > 10. Scale bars = 1 μm.

DOI: https://doi.org/10.7554/eLife.47918.023

The following source data and figure supplements are available for figure 5:

**Source data 1.** *Figure 4*_Intrinsic Ca.
DOI: https://doi.org/10.7554/eLife.47918.026
**Figure supplement 1.** Synaptic activity regulates postsynaptic DD spine density.
DOI: https://doi.org/10.7554/eLife.47918.024
**Figure supplement 2.** Ventral D-GABAergic motor neurons have dendritic spines.
DOI: https://doi.org/10.7554/eLife.47918.025

## Why study dendritic spines in *C. elegans*?

The prevalence of postsynaptic protrusions in vertebrate and invertebrate nervous systems suggests that spines are ancient structures and thus could be effectively investigated in a variety of model organisms (*Leiss, 2008*; *Petralia et al., 2016*). Our analyses revealed salient, conserved features of *C. elegans* dendritic spines: (1) A key role for the actin cytoskeleton in spine morphogenesis; (2) Postsynaptic receptor complexes at the tips of spines in close proximity to presynaptic active zones; (3) Postsynaptic calcium transients evoked by presynaptic activity and propagated from intracellular $Ca^{++}$ stores; (4) The presence of endoplasmic reticulum and ribosomes; (5) Regulation of spine density by presynaptic activity.

   *C. elegans* offers several advantages for studies of spine morphogenesis and function. Because *C. elegans* is transparent, live imaging does not require surgery or other invasive methods that are typically necessary for in vivo imaging of spines in an intact mammalian nervous system. Well-developed *C. elegans* genetic tools for targeted genomic manipulation (*Nance and Frøkjær-Jensen, 2019*) and unbiased forward genetic screens can be used to reveal new determinants of spine assembly. A recent study, for example, reported that neurexin, a conserved membrane protein and established regulator of synaptic assembly, is necessary for spine morphogenesis in DDs. Interestingly, in this case, neuroligin, the canonical neurexin ligand, is not required, suggesting a potentially new neurexin-dependent mechanism of synaptogenesis (*Oliver et al., 2018*; *Philbrook et al., 2018*).

   Our study confirmed that both DD and VD motor neurons display dendritic spines. Thus, other neurons reported to have 'short branches' in the original EM reconstruction of the adult *C. elegans* (*White et al., 1986*), the cholinergic (RMD, SMD) and GABAergic (RME) motor neurons and the interneuron (RIP), are likely to display *bona fide* dendritic spines. An ongoing effort to produce a gene expression fingerprint of each type of *C. elegans* neuron (*Hammarlund et al., 2018*; *Taylor, 2019*) may be useful for identifying genetic programs that uniquely correlate with spine morphogenesis since only a small number (*White et al., 1976*) of neurons have been reported to have spine-like structures. Finally, a developmentally regulated remodeling program (*Kurup and Jin, 2016*; *Petersen et al., 2011*; *White et al., 1978*) transforms presynaptic boutons into postsynaptic spines in larval DD neurons and thus could be especially useful for live imaging studies of synaptic plasticity and spine morphogenesis.

## Materials and methods

Plasmids used in this study:

| Plasmid name | Description | Cloned by |
|---|---|---|
| pACC4 | punc-25::LifeAct::GFP | Gateway cloning |
| pACC6 | pflp-13::LifeAct::GFP | InFusion cloning |
| pACC12 | pflp-13::LifeAct::mCherry | InFusion cloning |
| pACC22 | pflp-13::toca-1a::mCherry | InFusion cloning |
| pACC83 | pflp-13::GCaMP6s::SL2::mCherry | InFusion cloning |
| pACC86 | punc-4::ceChrimson::SL2::mCherry | InFusion cloning |
| pACC92 | punc-4::ceChrimson::SL2::3xNLS::GFP | InFusion cloning |
| pMLH09 | punc-25::gateway::GFP | Gift |
| pSH4 | pmyo-2::RFP | Gift |
| pSH21 | pstr-1::GFP | InFusion cloning |
| pSH40 | punc-4::HisCl::SL2::mCherry | Gift |
| pSH83 | pflp-13::miniSOG::SL2::BFP | InFusion cloning |
| pACC128 | pflp-13::myr::mRuby3 | InFusion cloning |

Strains used in this study:

| Strain name | Description | Reference |
|---|---|---|
| XMN46 | bgIs6 [pflp-13::mCherry; Pttx-3::RFP] II | (*Opperman and Grill, 2014*) |
| NC3315 | wdEx1016 [*pflp-13*::LifeAct::GFP; *pmyo-2*::RFP] | This study |
| NC3376 | ufls63 (pacr-2::RAB3::mCherry); wdEx1016[*pflp-13*::LifeAct::GFP; pmyo-2::RFP] | This study |
| NC3458 | *eri-1(mg366)*; wdEx1016 [*pflp-13*::LifeAct::GFP; *pmyo-2*::RFP] | This study |
| NC3355 | *toca-1(tm2056)* X; wdEx1016 [*pflp13*::LifeAct::GFP; *pmyo-2*::RFP] | This study |
| NC3357 | wdEx1029 [*pflp13*::LifeAct::GFP; *punc4*::HisCl::SL2::mCherry; *pmyo-2*::RFP] | This study |
| NC3455 | *ace-2(g72)* I; *ace-1*(p100) X; wdEx1016 [*pflp-13*::LifeAct::GFP; *pmyo-2*::RFP] | This study |
| NC3462 | *unc-17(e113)* IV; wdEx1016 [*pflp-13*::LifeAct::GFP; *pmyo-2*::RFP] | This study |
| NC3469 | wdEx1069 [*pflp13*::LifeAct::mCherry; *pstr-1*::GFP] | This study |
| NC3480 | wdEx1074 [*pflp-13*::miniSOG::SL2::BFP; *punc-25*::LifeActGFP; pmyo-2::RFP] | This study |
| NC3482 | *acr-12(ok367)* ufls126 [*pflp-13*::ACR12::GFP] X, wdEx1069 [*pflp-13*::LifeActmCherry; pstr-1::GFP] | This study |
| NC3484 | wdEx1078 [*pflp-13*::GCaMP6s::SL2::mCherry; pmyo-2::RFP] | This study |
| NC3486 | *unc-31(e169)* IV; wdEx1016 [*pflp-13*::LifeAct::GFP; *pmyo-2*::RFP] | This study |
| NC3489 | wdEx1083 [*punc-4*::ceChrimson::SL2::mCherry; pflp-13::LifeAct::GFP; *pmyo-2*::RFP] | This study |
| NC3569 | *lin-15(n765)*; wdIs117 [punc-4::ceChrimson::SL2::3xNLSGFP; lin-15+]; wdEx1112 [*pflp13*::GCaMP6s::SL2::mCherry] | This study |
| NC3608 | wdEx123[*pflp-13*::myr::mRuby; *pstr-1*::GFP] | This study |
| NC3609 | *toca-1(tm2056)* X; wdEx124[*pflp-13*::toca-1a::mCherry; *pflp-13*::LifeAct::GFP; *pmyo-2*::RFP] | This study |
| NC3506 | *lev-10(wd115)* I; wdEx1099[*pflp-13*::GFP1-10; *pflp-13*::LifeAct::mCherry; pmyo-2::RFP] | *He et al. (2019)* Genetics |
| NC3610 | *unc-68(r1162)* V; wdEx1112 [*pflp13*::GCaMP6s::SL2::mCherry] | This study |

## Worm breeding

Worms were maintained at 20°- 25°C using standard techniques (*Brenner, 1974*). Strains were maintained on NGM plates seeded with *E. coli* (OP-50) unless otherwise stated. The wild type (WT) is N2 and only hermaphrodite worms were used for this study. Staging as L3, early L4 (eL4), L4 and young adult worms was defined following vulva development as previously reported (*Chia et al., 2012*).

## Molecular biology

Gateway cloning was used to build pACC06 (*punc-25*::LifeAct::GFP). Briefly, plasmid pDONR221 (Plastino Lab) (*Havrylenko et al., 2015*) was used in the LR reaction with pMLH09 (*punc-25*::ccdB:: GFP) to create pACC06. Additional plasmids were created using InFusion cloning (Takara). The InFusion cloning module (SnapGene) was used to design primers to create the desired plasmid. Briefly, vector and insert fragments were amplified using CloneAmp HiFi polymerase. PCR products were gel-purified and incubated with In-Fusion enzyme for ligations. Constructs were transformed into

Stellar Competent cells and confirmed by sequencing (See full list of plasmids). Plasmids are available upon request. Addgene provided sequences for GCamP6s (#68119), and *C. elegans*-optimized Chrimson (ceChrimson, #66101) (*Schild and Glauser, 2015*). miniSOG sequence was a gift from the Jin lab (*Qi et al., 2012*). pSH40 was a gift from the Bargmann Lab (*Pokala et al., 2014*). *C. elegans*-optimized mRuby3 was a gift from Peri Kurshan (Shen Lab). TOCA-1a cDNA was a gift from Barth Grant (*Bai and Grant, 2015*). The 3X-NLS nuclear localization signal (pACC92) was a gift from Eviatar Yemini (Hobert Lab).

## Feeding RNAi

Clones from the RNAi feeding library (Source BioScience) were used in this study. RNAi plates were produced as described (*Petersen et al., 2011*). Briefly, RNAi bacteria were grown in the presence of ampicillin (50 µg/mL) and induced with IPTG (1 mM). 250 µL of the RNAi bacterial culture was seeded on NGM plates. RNAi plates were kept at 4°C for up to one week until used. RNAi experiments were set-up as follows: 3 to 5 L4 worms (NC3458) were placed on RNAi plates and maintained at 20°C. Four days later, F1 progeny was imaged as young adults.

## Electron microscopy

Young adult animals were fixed using high pressure freezing followed by freeze substitution, as previously described (*Mulcahy et al., 2018*; *Rostaing et al., 2004*), with minor modification: they were held at −90°C in acetone with 0.1% tannic acid and 0.5% glutaraldehyde for 4 days, exchanged with 2% osmium tetroxide in acetone, raised to −20°C over 14 hr, held at −20°C for 14 hr, then raised to 4°C over 4 hr before washing. Additional *en bloc* staining was performed with uranyl acetate for 2 hr at room temperature, followed by lead acetate at 60°C for 2 hr. Samples were embedded in Epon, cured at 60°C for 24 hr, then cut into 50nm-thick serial sections. Sections were not poststained. Images were taken on a FEI Tecnai 20 transmission electron microscope with a Gatan Orius digital camera, at 1 nm/pixel.

## 3D reconstruction

Images were aligned into a 3D volume and segmented using TrakEM2 (*Cardona et al., 2012*), a Fiji plugin (*Schindelin et al., 2012*). Neuron identity was assigned based on characteristic morphology, process placement, trajectory and connectivity (*Mulcahy et al., 2018*; *White et al., 1986*). The ventral and dorsal cord volumes contained the anterior-most 25 um of DD1, and 27 um of VD2, respectively. Volumetric reconstructions were exported to 3Ds Max for processing (3Ds Max, Autodesk).

## Airyscan microscopy

Worms were mounted on 10% agarose pads and immobilized with 15 mM levamisole/0.05% tricaine dissolved in M9. A Zeiss LSM880 microscope equipped with an Airyscan detector and a 63X/1.40 Plan-Apochromat oil objective lens was used to acquire super resolution images of the DD neuron. Images were acquired as a Z-stack (0.19 µm/step), spanning the total volume of the DD ventral process and submitted for Airyscan image processing using ZEN software. Developmental stage was determined by scoring gonad and vulva development (*Schindler and Sherwood, 2013*).

## Classification of spines

Spine shapes were determined from Z-projections of Airyscan images and by 3D-EM reconstruction. Mean and SD were determined using GraphPad. Spines were classified as thin/mushroom, filopodial, stubby or branched. Thin/mushroom spines displayed a constricted base (neck) and an expanded tip (head). Filopodial spines do not have a constricted base (no neck) but are protrusions of constant width. Stubby spines were recognized as protrusions with a wide base and tip. Branched spines were identified as protrusions with more than one visible tip.

## Ribosomal protein labeling in DD spines

To label ribosomes in DD spines, we used DD-specific RIBOS (*Noma et al., 2017*). To label DD spines we injected *Pflp-13*::LifeAct::mCherry plasmid into CZ20132 and used Airyscan imaging to examine transgenic animals (See Airyscan microscopy section).

## Actin dynamics

A Nikon microscope equipped with a Yokogawa CSU-X1 spinning disk head, Andor DU-897 EMCCD camera, high-speed piezo stage motor, 100X/1.49 Apo TIRF oil objective lens and a 1.5X magnification lens was used for live imaging. For measurements of LifeAct::GFP and cytosolic mCherry dynamics, L4 and young adults (NC3315 and XMN46) were mounted on 10% agarose pads and immobilized with 15 mM levamisole/0.05% tricaine dissolved in M9. Z-stacks (0.5 µm/step) were collected every 3 min. Movies were submitted to 3D-deconvolution on NIS-Elements using the Automatic algorithm and aligned with the NIS Elements alignment tool. For each movie, ROIs were defined along the dendritic shaft for each spine. Mean ROI Intensity was calculated for each time point and exported to Microsoft Excel. Background was determined from a neighboring region inside the worm and subtracted from the ROI in each timepoint. Mean intensity changes where normalized to the mean Intensity from the first timepoint of each movie. Intensity changes for LifeAct::GFP and mCherry were graphed using Prism6 software.

## GCaMP6s dynamics in DD spines

GCaMP6s imaging was performed on a Nikon microscope equipped with a Yokogawa CSU-X1 spinning disk head, Andor DU-897 EMCCD camera, high-speed piezo, 100X/1.49 Apo TIRF oil objective lens and a 1.5X magnification lens. NC3484 worms were immobilized using a combination of 3 µL of 100 mM muscimol (TOCRIS biosciences #0289) and 7 µL 0.05 µm polybeads (2.5% solids w/v, Polysciences, Inc #15913–10). Triggered acquisition was used to excite the GCaMP and mCherry signals with 488 nm and 561 nm lasers. Single plane movies were collected every second for at least 24 s. Movies were submitted for 2D-deconvolution on NIS-Elements using the Automatic algorithm. Movies collected with NIS-elements were aligned through time using the ND alignment tool. ROIs with the same area for each channel were defined in spines and on a neighboring region to determine background intensity for every time point. Mean ROI intensity was exported to Microsoft Excel for subtraction of mean fluorescence background intensity. Fluorescence at each timepoint was normalized to intensity at t = 0 for GCaMP6s and mCherry signals. Local peaks of GCaMP6s fluorescence were identified between neighboring neurons and the difference between the timepoints (deltaT) was calculated. Traces were graphed on Prism6.

To detect evoked calcium responses in DD neurons, NC3569 was grown for one generation on an OP50-seeded plate with freshly added ATR or carrier (EtOH). L4 worms were glued (Super Glue, The Gorilla Glue Company) to a microscope slide in 2 µL 0.05 µm polybeads (2.5% solids w/v, Polysciences, Inc #15913–10) plus 3 µL of M9 buffer and imaged under a coverslip.

GCaMP6s dynamics were recorded after all-spines activation on a Nikon microscope equipped with a Yokogawa CSU-X1 spinning disk head, Andor DU-897 EMCCD camera, high-speed piezo and 100X/1.49 Apo TIRF oil objective lens. Single plane images encompassing DD1 postsynaptic spines and adjacent VA and DA motor neurons were collected at two frames/second for 15 s. The sample was illuminated with a 561 nm laser at 2.5 s intervals (e.g., every five frames) for red light activation of ceChrimson expressed in cholinergic DA and VA motor neurons (*Punc-4*::ceChrimson::SL2::3xNLS::GFP) while maintaining constant illumination with a 488 nm laser to detect GCamP6s signals.

We used a sequential excitation/imaging protocol to measure $Ca^{++}$ transients in spines after local activation of cholinergic release. GCaMP6s signals were recorded after a single-spine activation on a Nikon A1R laser scanning confocal microscope using a 60X/1.4 N.A oil objective lens with Nyquist acquisition. A stimulation ROI was defined adjacent to a single DD1 or DD2 spine and illuminated with the 561 nm laser for less than 200 ms. GCaMP6s changes were recorded with 488 nm laser every 2 s for 30 s and 561 nm excitation was triggered after the 4th imaging time-point. Ryanodine-treated worms were soaked in 1 mM ryanodine (TOCRIS biosciences #1329) for 15 min before the imaging session and mounted on a slide using glue as above.

For quantifying GCaMP6s fluorescence, videos were aligned and 2D-deconvolved using NIS Elements software. Three regions of interest were defined to evaluate average fluorescence intensity from (1) a spine at the site of excitation; (2) an adjacent spine and (3) a distant spine, that is at least 9 µm away from the excitation point. Signal was collected from separate ROIs drawn on each spine and on a nearby region to capture background fluorescence. Mean fluorescence intensity of each ROI was exported into Excel for analysis. Background fluorescence was subtracted from each frame

and measurements were normalized to the initial time-point. Mean fluorescence traces were plotted using Prism6. Statistical analysis compared average GCaMP6s intensity of the first four frames vs GCaMP6s after stimulation ($5^{th}$ frame) using the Kruskal Wallis test for non-parametric samples and multiple comparisons.

### Temporal neuronal silencing with histamine chloride

Gravid adults were allowed to lay eggs for 2 hr on an OP50-seeded plate at 20°C to produce a synchronized population of L1 larvae. The middle time-point of the egg-laying session was considered $T_0$. At $T_{19}$ (time in hours), L1 larvae were transferred to control or histamine plates and maintained at 20°C until imaging on an LSM880 Airyscan microscope at the young adult stage. For control plates, 200 µL of water was added to OP-50 seeded NGM plates. For histamine plates, 200 µL of 0.5M Histamine, diluted in water, was added to OP-50 seeded NGM plates.

### Temporal neuronal activation

Gravid adults were allowed to lay eggs for 2 hr on an OP-50-seeded plate with freshly added ATR. The resultant synchronized population of L1 larvae was maintained at room temperature (23–25°C). At $T_{27}$ (L2 larvae), we used WormLab (MBF Bioscience) for exposure to repetitive cycles of 1 s ON + 4 s OFF for 6 hr of a 617 nm precision LED (Mightex PLS-0617-030-10-S). Images were collected on an LSM880 Airyscan microscope with 60X/1.4 Plan-Apochromat oil objective lens. Control worms were not exposed to light but grown on ATR plates. 100 mM ATR (Sigma, #A7410) was prepared in ethanol and stored at −20°C. 300 uM of ATR was added to OP-50 bacteria and seeded on NGM plates. Plates were dried in darkness overnight and used the next day for experiments.

### Image analysis

FIJI (*Schindelin et al., 2012*) and NIS-Elements software were used for data quantification. Z-stacks were flattened in a 2D projection and line scans were manually drawn along protrusions and perpendicular to the proximal shaft (*Figure 1d*) to determine the Protrusion/Shaft ratio (*Figure 1e*). Spine density was calculated using the counting tool in NIS Elements and then normalized to number of spines per 10 µm of dendrite length.

### Statistical analysis

For comparison between two groups, Student's T-test was used and $p < 0.05$ was considered significant. ANOVA was used to compare between three or more groups followed by Dunnett's multiple-comparison test. Standard Deviations between two samples were compared using an F-test and considered $p < 0.05$ as significant.

### Ablation of DD neurons

Twenty gravid adults (NC3480) were allowed to lay eggs for 2 hr with the middle time point considered $T_0$. At $T_{16}$, (hours) DD neurons were ablated by miniSOG (*Qi et al., 2012*) activation by exposing worms for 45 min to a 470 nm LED light (#M470L2, Thor Labs). Animals were then maintained at 20°C until imaging at ~$T_{60}$ as young adults.

## Acknowledgements

We thank M Francis and Alison Philbrook for sharing unpublished observations. We thank Douglas Holmyard (Nanoscale Biomedical Imaging Facility) for preparing EM samples and serial EM sections; Dylan Burnette, Aidan Fenix and Nilay Taneja for insightful imaging discussions; Peri Kurshan, Cori Bargmann, Yishi Jin, Barth Grant and Eviatar Yemini for reagents, and Erin Miller for help with aligning serial electron micrographs. Some *C. elegans* strains used in this work were provided by the Caenorhabditis Genetics Center, which is funded by the NIH Office of Research Infrastructure Programs (P40 OD010440). Airyscan and live-imaging data was acquired at the Vanderbilt Cell Imaging Shared Resource with help from Bryan Millis and Jenny Schafer (supported by NIH grants CA68485, DK20593, DK58404, DK59637 and EY08126). This work was supported by National Institutes of Health grants to DMM (R01NS081259 and R01NS106951), American Heart Association predoctoral

fellowships to ACC (18PRE33960581) and SP (19PRE34380582), National Science Foundation GRFP to SP (DGE:1445197) and a Canadian Institute of Health Research grant to MZ (FS154274).

## Additional information

### Funding

| Funder | Grant reference number | Author |
|---|---|---|
| National Institute of Neurological Disorders and Stroke | R01NS081259 | David M Miller III |
| National Institute of Neurological Disorders and Stroke | R01NS106951 | David M Miller III |
| American Heart Association | 18PRE33960581 | Andrea Cuentas-Condori |
| National Science Foundation | DGE:1445197 | Sierra Palumbos |
| Canadian Institutes of Health Research | FS154274 | Mei Zhen |
| American Heart Association | 19PRE34380582 | Sierra Palumbos |

The funders had no role in study design, data collection and interpretation, or the decision to submit the work for publication.

### Author contributions

Andrea Cuentas-Condori, Conceptualization, Resources, Data curation, Formal analysis, Validation, Investigation, Visualization, Methodology, Writing—original draft, Writing—review and editing; Ben Mulcahy, Resources, Data curation, Formal analysis, Validation, Investigation, Visualization, Methodology, Writing—review and editing; Siwei He, Resources, Visualization, Methodology, Writing—review and editing; Sierra Palumbos, Resources, Funding acquisition, Writing—review and editing; Mei Zhen, Resources, Supervision, Funding acquisition, Validation, Methodology, Project administration, Writing—review and editing; David M Miller III, Conceptualization, Resources, Supervision, Funding acquisition, Validation, Methodology, Project administration, Writing—review and editing

### Author ORCIDs

Andrea Cuentas-Condori (iD) http://orcid.org/0000-0002-4847-0031
Ben Mulcahy (iD) http://orcid.org/0000-0002-3336-245X
Sierra Palumbos (iD) https://orcid.org/0000-0002-3595-984X
Mei Zhen (iD) http://orcid.org/0000-0003-0086-9622
David M Miller III (iD) https://orcid.org/0000-0001-9048-873X

### Decision letter and Author response

Decision letter https://doi.org/10.7554/eLife.47918.029
Author response https://doi.org/10.7554/eLife.47918.030

## Additional files

### Supplementary files

• Transparent reporting form DOI: https://doi.org/10.7554/eLife.47918.027

### Data availability

All data generated or analysed during this study are included in the manuscript and supporting files. Source data files have been provided for Figures 1, 4 and 5.

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
