## [Decision Letter]

Thank you for submitting your article "*C. elegans* neurons have functional dendritic spines" for consideration by *eLife*. Your article has been reviewed by three peer reviewers, one of whom is a member of our Board of Reviewing Editors, and the evaluation has been overseen by Eve Marder as the Senior Editor. The reviewers have opted to remain anonymous.

The reviewers have discussed the reviews with one another and the Reviewing Editor has drafted this decision to help you prepare a revised submission.

Summary:

All three reviewers agreed that this work had gone beyond previously published works and systematically characterized dendritic spines in the worms. All three reviewers agree that these results will form the basis of further genetic analyses that might bring more genetic and molecular understanding of spines.

Essential revisions:

1) One of the most important functions of spines is to isolate membrane domain and cytoplasmic space for localized signaling. The manuscript states that the Ca^++^ signals observed can be both local and global. The reviewers are interested in testing whether there are local Ca^++^ signals. There are several different ways to do this. The authors just need to do one of these experiments if possible. For example, a local optical stimulation with ChR2 expressing axon to see if spine specific localized Ca^++^ increases can be triggered. Alternatively, the author might consider during more detailed analyses of their spontaneous Ca^++^ imaging to show that local Ca^++^ events and use mutants or pharmacology to test if there are difference between the local and global Ca^++^ signals. At the very least, the authors should show examples of local events and provide more discussion about local vs. global signals considering the specific features of *C. elegans* neurons.

2) The authors should also provide a discussion on the point that worm spines do not contain clear postsynaptic density by EM which is a key feature of the vertebrate spines.

3) Some of the best images of the spines are seen with transgenic worms expressing LifeAct::GFP, which binds and labels actin cables. Actin is a dynamic structure that can be influenced by the presence of actin-binding proteins. Is it possible that the transgenic LifeAct::GFP, through its binding and possible modulation of actin, is actually promoting or altering the structure and/or number of spines? Is the reporter itself creating something artificial? For example, it would be great if the authors could show that introduction of the LifeAct::GFP transgene into worms expressing mCherry did not alter the number or quantified morphology of spines in LifeAct::GFP mCherry double transgenic worms compared to worms that only express the mCherry reporter alone.

Along the same line, actin regulators such as toca-1, arp2.3, wave complex are likely to have pleiotropic phenotypes. Cell autonomy experiments in a mutant background would provide confidence that actin is required for the spines. The authors don't need to do all of the mutants.

---

## [Author Response]

Essential revisions:1) One of the most important functions of spines is to isolate membrane domain and cytoplasmic space for localized signaling. The manuscript states that the Ca^++^ signals observed can be both local and global. The reviewers are interested in testing whether there are local Ca^++^ signals. There are several different ways to do this. The authors just need to do one of these experiments if possible. For example, a local optical stimulation with ChR2 expressing axon to see if spine specific localized Ca^++^ increases can be triggered. Alternatively, the author might consider during more detailed analyses of their spontaneous Ca^++^ imaging to show that local Ca^++^ events and use mutants or pharmacology to test if there are difference between the local and global Ca^++^ signals. At the very least, the authors should show examples of local events and provide more discussion about local vs. global signals considering the specific features of *C. elegans* neurons.

We thank reviewers for suggesting the further evaluation of calcium transients in DD spines. We have addressed this point as follows:

First, we targeted single spines for activation (L4 stage) and recorded Ca^++^ changes in three different regions of interest: (1) at the excited spine; (2) at an adjacent spine; and (3) at a distant spine (Figure 4I). Activation was achieved by optogenetic stimulation of Chrimson in a presynaptic cholinergic domain at the tip of a DD spine; Ca^++^ was measured with GCaMP6s in postsynaptic spines. In our microscope (Nikon A1R), optogenetic activation of Chrimson in a restricted ROI required a sequential activation/imaging protocol such that the GCaMP6s signal was captured ~2 seconds after Chrimson activation (Figure 4J) (see below for technical note). In this experimental paradigm, we detected elevated Ca^++^ in all three classes of spines (i.e. excited, adjacent and distant). We interpret these findings to mean that local activation of a spine likely spreads to nearby spines within the 2 second interval between Chrimson activation and GCaMPs imaging in our experimental paradigm.

Because similar experiments in developing hippocampal neurons have shown that Ca^++^ release from intracellular stores is required for propagating a local Ca^++^ signal to adjacent spines (Lee et al., 2016), we repeated this experiment with ryanodine to block Ca^++^ release from intracellular stores. In this case, we observed no significant changes in Ca^++^ levels in spines (Figure 4K), suggesting that the propagation of Ca^++^ waves among DD spines depends on intracellular stores. Consistent with this finding, we observed SER-like structures throughout DD spines and dendritic shaft (Figure 3A-B). Additionally, a mutation that disrupts the UNC-68/Ryanodine receptor, dramatically reduces intrinsic Ca^++^ waves in spines (Figure 4—figure supplement 1A). Overall, these results are consistent with a model in which local spine stimulation is propagated to neighboring spines by Ca^++^ release from internal stores.

We acknowledge that the interpretation of our results is indirect; that local activation of a spine spreads to adjacent spines during the ~2 sec interval between Chrimson activation and GCaMP6s imaging. If this model is correct, then it should be possible to detect a spreading Ca^++^ wave directly by either simultaneous activation/ imaging or at least with a much shorter dead time (i.e. << 2 sec) in our set up. Although technically possible, the cost of upgrading (Miniscanner or Digital Micromotor Device + LED) one of our Nikon microscopes (A1R or spinning disk) for either of these imaging modalities would cost ~$30,000 which is currently not a feasible solution.

2) The authors should also provide a discussion on the point that worm spines do not contain clear postsynaptic density by EM which is a key feature of the vertebrate spines.

Thanks for this suggestion. We have now discussed the absence of postsynaptic densities in electron micrographs of *C. elegans* spines (subsection “Dendritic spines of DD neurons directly appose presynaptic terminals”, last paragraph). We note that although postsynaptic densities (PSDs) are commonly observed in vertebrate spines, this effect is correlated with neurotransmitter type: EMs of glutamatergic synapses show prominent PSDs (Ziff, 1997) but detectable PSDs are either absent or much less robust for glycineric, GABAergic and cholinergic synapses (Knott et al., 2002; Kubota et al., 2007; Umbriaco et al., 1994).

3) Some of the best images of the spines are seen with transgenic worms expressing LifeAct::GFP, which binds and labels actin cables. Actin is a dynamic structure that can be influenced by the presence of actin-binding proteins. Is it possible that the transgenic LifeAct::GFP, through its binding and possible modulation of actin, is actually promoting or altering the structure and/or number of spines? Is the reporter itself creating something artificial? For example, it would be great if the authors could show that introduction of the LifeAct::GFP transgene into worms expressing mCherry did not alter the number or quantified morphology of spines in LifeAct::GFP mCherry double transgenic worms compared to worms that only express the mCherry reporter alone.

To address the possibility of LifeAct-dependent effects on spine morphology, we performed independent imaging experiments with a membrane marker, myristolated mRuby (MYR::mRuby) (Figure 1—figure supplement 1). This experiment determined that spine density does not differ between LifeAct::GFP vs. MYR::mRuby-expressing DD neurons (Figure 1—figure supplement 1A) and that the “thin/mushroom” class of spines predominates in both cases. We did detect some differences in the distribution of specific spine morphologies. The most significant difference is for stubby spines, which occur at a higher frequency with the MYR::mRuby label (Figure 1—figure supplement 1C). To summarize the significance of these findings we state: “These differences could reflect the relative ease of scoring different spine morphologies with markers for either the cell membrane (MYR::mRuby) or actin cytoskeleton (LifeAct::GFP). Alternatively, over-expression of these labels could alter spine morphology but, in this case, does not appear to perturb overall spine density”.

Along the same line, actin regulators such as toca-1, arp2.3, wave complex are likely to have pleiotropic phenotypes. Cell autonomy experiments in a mutant background would provide confidence that actin is required for the spines. The authors don't need to do all of the mutants.

Thank you for this suggestion. To address this point, we showed that expression of TOCA-1 in DD neurons rescues the spine-density defect of a *toca-1* mutant thus confirming that TOCA-1 acts cell autonomously in DD neurons to regulate spine density. (Figure 1—figure supplement 2G, subsection “DD spines are shaped by a dynamic actin cytoskeleton”, last paragraph).